# OpenReview forum: "BenchOverflow: Measuring Overflow in Large Language Models via Plain-Text Prompts"
_TMLR — Accepted by TMLR_

### Review · Reviewer_QatD · 2025-10-16

**Summary Of Contributions:**

This work investigates a specific failure mode of LLMs, where simple/plain natural language prompts lead to excessive generation – defined as 5000 tokens in the context of this work. Such overgeneration may lead to issues such as latency, denial-of-wallet and waste of resources. To demonstrate this issue, this paper introduces a benchmark – BenchOverflow – consisting of 9 families of prompting strategies where they show that multiple families of models (both open and closed source) suffer from this issue. Additionally, they suggest a simple approach to remediate this problem with a simple prompting fix.

*Strengths*:
- The paper is easy to read and clearly distinguishes itself from other works: in this work authors focus on plain/natural prompts, instead of jailbreaks or special tokens.
- This work addresses a very relevant problem: as LLMs are increasingly integrated into different systems and applications, measuring the length of generation is a very important aspect of making systems reliable. It is even more important to ensure robustness to simple attacks that could cause latency or denial-of-wallet.
- Experiments are conducted using a broad set of recent and popular models that show this issue exists across families: from closed sources to open sources.
- The benchmark contains a broad set of prompts, divided into 9 different families. Prompts are organized under a taxonomy that makes it possible for a model to generate multiple prompts.
- The analysis to assess the severity of the problem is carried out through clear statistical evidence: empirical cumulative distribution functions to measure central tendency and tail behavior, cap-saturation rates to quantify generation at certain thresholds, and consistency analysis to assess the actual stability of overflow across families of models. Each prompt is evaluated in independent runs which make the results reliable.
- Interesting analysis of model consistency: sometimes there is refusal from the LLM (some sort of declination from the model) and yet, long generation happens. A deeper dive into this problem would be interesting to make LLM more self-consistent.

*Weaknesses*:
- The proposed defensive mechanism seems weak and would not guarantee that the problem is clearly resolved. More imporantly, it is surprising that the overall experimentation is done across many models, with good statistical significance and yet the proposed mechanism to mitigate the problem is carried out only for two models.
- While the problem of excessive generation is clearly described, the assumption of possible denial-of-wallet, while intuitive, is not sufficiently supported with evidence. 5000 tokens do not seem overly difficult to serve, in fact authors do not report any issue even when the LLMs were generating at or beyond 5,000 tokens. The overall problem presented in this paper, while intuitively relevant, seems a bit overstated.
- Some of the prompts (for instance, “generate 1,200 unique trivia questions...”) are so explicit that it’s hard to blame the model, especially if 5,000 tokens is an acceptable threshold to serve. It would be interesting to see whether more casual prompts could lead to over-generation.
- The work would benefit from a more precise framework for defining length control. Some user requests legitimately require long outputs, and it is unclear what behavior should be considered “correct” in those cases.

**Additional Comments:**

N/A

**Audience:**

Yes

**Audience Explanation:**

This work presents a problem that is relevant for the community - both for researcher working on alignment but in general for developers that need to ensure applications do not suffer from issues such as latency or denial-of-wallet.

**Claims And Evidence:**

Yes

**Claims Explanation:**

The submission is supported by accurate experiments but some of the claims and issues reported are a bit overstated.

**Requested Changes:**

- Test the defensive mechanism for all models, adopting the same statistical rigor as in the main experiment
- Either change the tone to the severity of the problem or show examples as proof that such denial-of-wallet and high latency are so easy to obtain. The problem of over-generation from LLMs is something that researchers working on alignment should consider, but some of the claim seem overstated.
- A deeper analysis on models answering with refusal but over-generating would be interesting to improve consistency of LLMs.

---

### Review · Reviewer_mK8B · 2025-11-27

**Summary Of Contributions:**

This paper introduces BenchOverflow, a benchmark for studying “Overflow” in LLMs: cases where benign-looking plain-text prompts elicit excessively long outputs. The authors define a taxonomy of nine overflow-inducing prompting strategies (e.g., explicit forced length, tokenizer stress, infinite generation, quote, recursive details) and create >300 prompts per strategy using a meta-prompting pipeline with positive and negative in-context examples. They then evaluate 9 instruction-tuned LLMs (6 open, 3 closed) under a uniform 5k token generation budget, measuring length distributions, cap saturation rates (CSR@1k/3k/5k), within-prompt variance, and cross-model correlations. They also classify refusals with an LLM-as-judge and study a simple defense: a fixed conciseness reminder, which usually shifts distributions left and reduces CSR but does not fully neutralize the strongest vectors. The authors release the dataset and describe ethical and reproducibility considerations.

**Audience:**

Yes

**Audience Explanation:**

TMLR’s audience includes researchers and practitioners who care about the reliability, safety, and efficiency of large language models. This paper squarely targets all three:

- It frames “Overflow” as a reliability and safety issue (unbounded generation, denial-of-wallet / cost amplification, DoS-like behavior) rather than a cosmetic style problem. That aligns well with current interests in robust deployment of LLMs.

- It introduces a concrete, reusable benchmark (BenchOverflow) and taxonomy of overflow-inducing prompt strategies, evaluated systematically across multiple open and closed models. This is exactly the kind of empirical, reproducible resource that TMLR readers use for follow-up work and model comparisons.

- It speaks directly to practical deployment concerns (token budgets, cap saturation rates, simple prompt-level defenses) that matter to both academic groups running shared clusters and industry teams managing inference cost and latency.

**Claims And Evidence:**

No

**Claims Explanation:**

- The paper claims that plain-text prompts can systematically induce very long generations across models and that this varies by strategy and model family. This is backed by well-documented experiments on 9 instruction-tuned LLMs, multiple runs per prompt, and clear aggregate statistics (length distributions, CSR@1k/3k/5k, within-prompt variance, and cross-model correlations). The plots and tables are consistent with the narrative and support the conclusions.

- The claim that certain strategies (e.g., explicit forced length, tokenizer stress, infinite generation) are particularly effective at saturating token caps is strongly supported by the CSR analysis and per-strategy breakdowns.

- The claim that a simple conciseness reminder can reduce overflow but not fully neutralize it is supported by before/after comparisons on both BenchOverflow and a benign prompt set (at least in terms of length statistics)

**Requested Changes:**

1. Threat model and “benign” status of prompts

The paper emphasizes that prompts are “benign, plain-text” and not jailbreaks or adversarial suffixes. However, many examples (e.g., “write an infinite poem…”, “manually compute the sum 1..1,000,000 showing each intermediate step,” “reproduce the full text of Common Sense,” “reformat the UN Charter as a detailed musical score”) are arguably adversarial or at least deliberately pathological from a user-intent perspective. It would help to clarify **the intended deployment scenarios** where such prompts are likely to occur organically (e.g., student homework misuse, stress-testing by curious users, or targeted cost-amplification attacks). Right now, the claim that these are “benign” could be softened or better justified with empirical evidence from real-world logs or anecdotal reports.

2. Add utility/cost tradeoff

The evaluation focuses purely on output length, CSR, and stability. There is no measurement of task utility, user satisfaction, or success rate under the defense vs baseline, even on the benign OASST2 prompts.As a result, we cannot tell whether the conciseness reminder meaningfully harms normal task performance or user experience, which is crucial if the paper wants to argue for length control as a deployable reliability control. Even a small additional experiment would help: apply the conciseness reminder to OASST2 prompts and report human or LLM-as-judge scores on informativeness/adequacy vs the original outputs. Alternatively, piggyback on existing automatic metrics where applicable. This would make the defense section significantly stronger.

3. LLM-as-judge reliability

At minimum, add a short note in Limitations acknowledging that refusal labels are not human-validated and may contain errors, and that future work should calibrate this step with human annotation. If possible, a small spot-check (e.g., 100 samples) would greatly strengthen the credibility of the refusal analysis.

---

### Review · Reviewer_RQRE · 2025-12-10

**Summary Of Contributions:**

This paper investigates 'overflow failures' in LLMs—a vulnerability where models generate excessive, unintended output sequences that can lead to service disruptions. To study this, the authors introduce BenchOverflow, a benchmark consisting of nine distinct scenarios designed to provoke long, uncontrolled output sequences. Using this benchmark, the paper evaluates the robustness of several top-performing open-source and closed-source LLMs. Finally, the authors propose and test a simple defense mechanism to mitigate these overflow issues and report its effectiveness on the new benchmark

**Audience:**

Yes

**Audience Explanation:**

This paper investigates a critical aspect of LLM deployment. It is important to ensure that such deployments are resilient against adversarial prompts, specifically within the nine scenarios identified, where prompt alteration can trigger failures. Consequently, this paper is of clear interest to researchers and practitioners working on LLM safety, robustness, and deployment efficiency

**Claims And Evidence:**

Yes

**Claims Explanation:**

The claims in the submission are well-supported by clear evidence. The authors identify 9 distinct scenarios susceptible to overflow failures and construct a carefully curated dataset to test them. The empirical evaluation effectively demonstrates that the proposed BenchOverflow benchmark triggers excessive sequence generation across a variety of existing LLMs. Furthermore, the experiments substantiate the claim that the proposed simple defense is effective, showing a demonstrable reduction in overflow incidents

**Requested Changes:**

1. While the newly added Figure 1 clarifies the general workflow,  some critical details could be added. It is unclear how the BenchOverflow is curated. Which model and prompts did you use to generate those prompts in meta prompting process?
2. While Table 1 have provides examples for each category,  it will be beneficial to provide more examples in the appendix as the BenchOverflow is not yet released (please correct me if I am wrong).
3. In Figure 5 a, the distribution of STD of sequence lengths. To my understanding, the choice of temperature (temperature =1 is a reasonable choice) could fluctuate the generated sequence by a large margin. Also, each model operates under a different default temperature. I feel this is also a major limitation of the existing experiment that should be included in the limitations section

---

### Decision · Action_Editor_duUf · 2026-01-10

**Recommendation:** Accept as is

**Audience:**

Yes

**Audience Explanation:**

The paper addresses practical LLM deployment concerns—cost, latency, and service reliability—that are directly relevant to TMLR's audience of researchers and practitioners. It provides a reproducible benchmark for measuring length-control robustness across models, filling a gap in systematic evaluation of overflow risks. The focus on deployment safety and the reusable evaluation protocol make this work valuable for both academic research and production systems.

**Claims And Evidence:**

Yes

**Claims Explanation:**

The claims are well-supported through systematic experiments across 9 LLMs with multiple runs per prompt, clear statistical metrics (CSR, ECDFs, variance analysis), and consistent results. The authors provide comprehensive evidence for overflow susceptibility across model families, effectiveness of the conciseness reminder defense, and refusal-versus-overflow behavior. Methodological details are transparent, examples are provided, and limitations (temperature sensitivity, LLM-as-judge annotation) are explicitly acknowledged.